# Molecular Characterization and Functional Analysis of Hypoxia-Responsive Factor Prolyl Hydroxylase Domain 2 in Mandarin Fish (*Siniperca chuatsi*)

**DOI:** 10.3390/ani13091556

**Published:** 2023-05-06

**Authors:** Yang Yu, Jian He, Wenhui Liu, Zhimin Li, Shaoping Weng, Jianguo He, Changjun Guo

**Affiliations:** 1State Key Laboratory for Biocontrol, Southern Marine Science and Engineering Guangdong Laboratory (Zhuhai), Guangdong Provincial Key Laboratory of Marine Resources and Coastal Engineering, Guangdong Provincial Observation and Research Station for Marine Ranching of the Lingdingyang Bay, School of Marine Sciences, Sun Yat-sen University, 135 Xingang Road West, Guangzhou 510275, China; 2Guangdong Province Key Laboratory for Aquatic Economic Animals, School of Life Sciences, Sun Yat-sen University, 135 Xingang Road West, Guangzhou 510275, China

**Keywords:** mandarin fish, HIF-1α, hypoxia, PHD2

## Abstract

**Simple Summary:**

Hypoxic stress often occurs in aquaculture environments and is primarily mediated by the hypoxia-inducible factor 1 (HIF-1) signaling pathway. Prolyl hydroxylase domain proteins (PHD) are cellular oxygen-sensing molecules that regulate the stability of HIF-1α. In this study, the characterization of the PHD2 from mandarin fish *Siniperca chuatsi* (*sc*PHD2) and its roles in the HIF-1 signaling pathway were investigated. This study furthers our understanding of the molecular mechanisms underlying hypoxia adaptation in teleost fish.

**Abstract:**

With increased breeding density, the phenomenon of hypoxia gradually increases in aquaculture. Hypoxia is primarily mediated by the hypoxia-inducible factor 1 (HIF-1) signaling pathway. Prolyl hydroxylase domain proteins (PHD) are cellular oxygen-sensing molecules that regulate the stability of HIF-1α through hydroxylation. In this study, the characterization of the PHD2 from mandarin fish *Siniperca chuatsi* (*sc*PHD2) and its roles in the HIF-1 signaling pathway were investigated. Bioinformation analysis showed that *sc*PHD2 had the conserved prolyl 4-hydroxylase alpha subunit homolog domains at its C-terminal and was more closely related to other Perciformes PHD2 than other PHD2. Tissue-distribution results revealed that *scphd2* gene was expressed in all tissues tested and more highly expressed in blood and liver than in other tested tissues. Dual-luciferase reporter gene and RT-qPCR assays showed that *sc*PHD2 overexpression could significantly inhibit the HIF-1 signaling pathway. Co-immunoprecipitation analysis showed that *sc*PHD2 could interact with *sc*HIF-1α. Protein degradation experiment results suggested that *sc*PHD2 could promote *sc*HIF-1α degradation through the proteasome degradation pathway. This study advances our understanding of how the HIF-1 signaling pathway is regulated by *sc*PHD2 and will help in understanding the molecular mechanisms underlying hypoxia adaptation in teleost fish.

## 1. Introduction

Oxygen is an essential element for the development, survival, and normal functions of all metazoans [1,2]. A low level of oxygen or hypoxia is a common physiological and pathological phenomenon that can elicit stress responses and even inflict damage to organisms [3]. To adapt to hypoxia stress, organisms have formed a series of regulatory mechanisms, such as regulating some genes and proteins through oxygen receptors and signal transduction pathways in organisms [4]. In various regulatory pathways, the hypoxia-inducible factor 1 (HIF-1) signaling pathway is the most important and most commonly studied regulatory pathway because it plays a crucial regulatory role in oxygen consumption and delivery [5,6]. In the HIF-1 signaling pathway, prolyl hydroxylase domain proteins (PHD) are particularly important to regulate HIF-1 stability [7,8].

PHDs were originally identified as catalytic oxidation enzymes and primarily revealed to hydroxylate HIF-α [9]. The activities of these enzymes depend on oxygen concentration and iron content because they use molecular oxygen as a substrate and ferrous ion as a cofactor [10]. PHDs have three homologs, including PHD1, PHD2, and PHD3, which belong to the Fe(II) and 2-oxoglutarate-dependent dioxygenase family [11]. PHDs are evolutionarily conservative and have a similar conserved C-terminal catalytic domain [12]. In addition to a C-terminal PHD, PHD2 contains an N-terminal myeloid zinc finger structure (ZF-MYND) compared with PHD1 and PHD3. This zinc finger is absent in PHD1 and PHD3 during the process of evolution [13,14]. Under normoxia condition, the PHDs hydroxylate two proline residues (Pro402/564 site) in the oxygen-dependent degradation domain of HIF-1α, and then von Hippel-Lindau tumor suppressor protein recognizes this, leading to HIF-1α degradation through the ubiquitin-proteasome pathway [15,16,17,18,19,20]. However, under hypoxia, the catalytic activity of PHDs is inhibited by the lack of oxygen, so HIF-1α is not hydroxylated and accumulated [21,22]. HIF-1α and HIF-1β then combine to form a heterodimer that could bind to its downstream hypoxia response element (HRE) to initiate the transcription of downstream genes, such as *lactate dehydrogenase A* (*ldha*), *glucose transporter-1* (*glut-1*), and *vascular endothelial growth factor* (*vegf*) [23,24,25,26,27]. Therefore, HIFs and PHDs are critical mediators in the adaptive response to hypoxia [28,29].

The mandarin fish, *Siniperca chuatsi*, is an aquaculture organism with high economic value in China’s aquaculture market [30,31]. Increasing market demand for mandarin fish also expand its value. The excessively high breeding density and the excessive bait feeding amount led to hypoxia [32]. Aquatic animals living in hypoxic environments must adapt to this condition to survive, and many important types of economic fish are highly intolerant of a low-oxygen environment, causing huge economic losses to aquaculture [33,34]. To understand the mechanism of PHD2 regulating HIF response to hypoxia in mandarin fish, PHD2 from mandarin fish was characterized and its function in the HIF-1 signaling pathway was investigated for the first time. This work can help subsequent research on the molecular mechanism of hypoxia adaptation of mandarin fish.

## 2. Materials and Methods

### 2.1. Fish and Cells

Fifteen healthy mandarin fish (body weight of 75–100 g) were purchased from a farm in Guangdong province. They were bred in a laboratory recirculating fresh-water system for 2 weeks to acclimatize, and the water temperature was maintained at 27 °C. Fish were anaesthetized with MS-222 (40 mg/L, Sigma–Aldrich, St. Louis, MO, USA) for tissue sampling. All animal experiments were performed in accordance with the regulations for animal experimentation of Guangdong Province, China and permitted by the Ethics Committee of Sun Yat-sen University (no. 2019121705). Mandarin fish fry (MFF-1) cell line was constructed and maintained in our laboratory, it was cultured in Dulbecco’s modified Engle’s medium (Gibco, Grand Island, NY, USA) supplemented with 10% fetal bovine serum (Gibco) at 27 °C in a moist atmosphere containing 5% CO_2_ [35]. Cells transfection was conducted with Transfect EZ3000 Plus (eLGbio, Guangzhou, China) according to the manufacturer’s instructions. Prior to transfection, cells were directly seeded in different culture plates according to different experimental requirements. Cells were transfected using EZ3000 Plus in a serum-free culture medium (Opti-MEM, Gibco).

### 2.2. Molecular Cloning of Mandarin Fish PHD2 (scPHD2) cDNAs

The *sc*PHD2 sequence was obtained from the transcriptome data (data unpublished). To amplify the *sc*PHD2, random amplification of cDNA ends RACE-PCR was performed according to the manufacturer’s instructions and the primers used in this cloning are shown in Table 1. Total RNAs of the MFF-1 cells were isolated by using Trizol reagent (Thermo Fisher Scientific, Waltham, MA, USA) according to the instructions, followed by treatment with RNase-free DNase (Promega, Madison, WI, USA) to remove contaminating DNA. cDNAs were synthesized from 1 μg of total RNAs with HiScript^®^ III 1st-Strand cDNA Synthesis Kit (Promega) following the manufacturer’s instruction. cDNAs were used as templates for nested PCR reactions. The PCR amplification was performed under the following conditions: 1 cycle of 95 °C for 5 min, 30 cycles of 95 °C for 30 s, 55 °C for 30 s, and 72 °C for 2 min, with an additional elongation at 72 °C for 10 min after the last cycle. Finally, the PCR products were purified, cloned into the pMD18-T vector (Takara, Tokyo, Japan), and sequenced (Tsingke, Beijing, China).

### 2.3. Sequence Analysis

Homology proteins of PHD were collected at the National Center for Biotechnology Information (http://www.ncbi.nlm.nih.gov/blast (accessed on 10 December 2022). The predicted amino acid sequence of *sc*PHD2 was analyzed using the Simple Modular Architecture Research Tool (SMART) program (http://smart.embl-heidelberg.de/ (accessed on 10 December 2022). Multiple sequence alignments were generated using the Clustal X v2.0 program and annotated using GeneDoc v 2.7.0 software. The phylogenetic tree of PHD sequences was constructed according to the alignment of amino acid sequences through the neighbor-joining method using the Molecular Evolutionary Genetics Analysis (MEGA) v10.0 program, with 1000 bootstrap replicates.

### 2.4. Three-Dimensional Structure Prediction

*sc*PHD2 protein structure was predicted by applying the homology modeling technique in Alphafold v2.3.0. The monomer_casp14 model was used for structure prediction at default parameters, and the following database versions were used: values of pdb_mmcif, pdb_seqres, uniport, and uniref90 (accessed on 14 December 2022) [36,37]. The predicted local-distance difference test (pLDDT) measured the confidence degree of the structure prediction, providing a better metric for identifying ordered and disordered regions.

### 2.5. Dual-Luciferase Reporter Gene Assays

The role of *sc*PHD2 in the HIF-1 signaling pathway was investigated by dual-luciferase reporter gene assays. When the MFF-1 cells reached approximately 70% confluency, they were used for transfection. Four treatments were designed: 0.2 μg pGL4-HREs-luc (Promega) + 0.2 μg pCMV-Flag (Takara) + 0.2 μg pCMV-Myc (Takara) + 0.02 μg pRT-TK (Promega); 0.2 μg pGL4-HREs-luc + 0.2 μg Flag-*sc*HIF-1α + 0.2 μg pCMV-Myc + 0.02 μg pRT-TK; 0.2 μg pGL4-HREs-luc + 0.2 μg Flag-*sc*HIF-1α + 0.2 μg Myc-*sc*PHD2 + 0.02 μg pRT-TK; 0.2 μg pGL4-HREs-luc + 0.2 μg pCMV-Flag + 0.2 μg Myc-*sc*PHD2 + 0.02 μg pRT-TK. At 24 h post-transfection, the firefly luciferase and *Renilla* luciferase (as control) were determined. The total cell lysates were conducted with a Dual-Luciferase Reporter Gene Assay Kit (Promega) in accordance with the manufacturer’s instructions. Luciferase activity was measured in Glomax (Promega). All experiments were performed independently at least three times with three technical replicates for each experiment.

### 2.6. Real-Time Quantitative PCR (RT-qPCR)

To determine the tissue-specific expression analysis of *scphd2* and levels of HIF-1 signaling pathway downstream genes, the expression levels of genes were measured using RT-qPCR. The RT-qPCRs were performed with SYBR^®^ premix ExTaq^TM^ (Takara) on a LightCycler 488 instrument (Roche Diagnostics, Switzerland). Primers for RT-qPCR were designed using Primer Express v3.0 software (Applied Biosystems) (Table 2). For tissue distribution analysis of *scphd2*, total RNAs from different tissues that were subsequently reverse transcribed were prepared as previously described [38]. The expression levels of *scphd2*, *scvegf*, *scglut1* and *scldha* were detected using the corresponding quantitative PCR forward and reverse primers. RT-qPCRs were performed in triplicate. RT-qPCRs were run at 95 °C for 5 s, at 60 °C for 40 s, and 70 °C for 1 s, followed by 40 cycles. Reactions were performed in triplicate and analyzed individually, relative to *β-actin* gene (an internal housekeeping control). The RT-qPCR data of target genes were analyzed using the Q-gene statistics add-in followed by unpaired sample *t*-test.

### 2.7. Indirect Immunofluorescence Assay (IFA)

For subcellular localization analysis, IFA was used to observe the proteins’ subcellular localization. Endotoxin-free plasmids of Flag-*sc*HIF-1α and Myc-*sc*PHD2 were transfected into MFF-1 cells. After 48 h post-transfection, the cells were washed with PBS buffer (pH 7.4) three times, fixed with anhydrous methanol for 15 min at −20 °C, and permeabilized using 0.5% Triton X-100 for 10 min. After blocking with 5% normal goat serum (Boster, Wuhan, China) in PBS for 1 h, and incubated with anti-Myc or anti-Flag tag antibodies (Sigma-Aldrich). Antibody binding was detected using the antibody conjugated with Alexa Fluor 488 or 594 (Thermo Fisher). Hoechst 33342 (Thermo Fisher) was used for nuclear staining. Images were obtained using a fluorescence microscope (Zeiss LSM510, Carl Zeiss AG, Oberkochen, Germany).

### 2.8. Co-Immunoprecipitation (Co-IP) and Western Blot Analysis

Endotoxin-free plasmids of Flag-*sc*HIF-1α and Myc-*sc*PHD2 were transfected into MFF-1 cells, and pCMV-Flag or pCMV-Myc in MFF-1 cells served as the control group. At 48 h post-transfection, the cells were washed with PBS, lysed with lysis buffer (Beyotime, Shanghai, China) containing a cocktail protease inhibitor (Merck Millipore, Billerica, MA, USA), and incubated on ice for 30 min. After centrifugation for 10 min at 12,000× *g*, supernatants were collected and treated with a Pierce™ c-Myc-Tag Magnetic IP/Co-IP Kit or a Pierce™ Flag-Tag Magnetic IP/Co-IP Kit (Thermo Fisher) in accordance with the manufacturer’s instructions. Then, the protein samples were separated by 10% SDS-PAGE and transferred onto nitrocellulose membranes (GE Healthcare Biosciences, Pittsburgh, PA, USA). The membranes were blocked in 5% skim milk in PBST buffer (PBS with 0.1% Tween 20) at room temperature for 1 h. After washing the membranes three times with PBST for 10 min each time, they were incubated with the appropriate primary and secondary antibodies at room temperature for 2 h. Following another extensive washing, protein bands were visualized using a High-Sig Chemiluminescence (ECL) Western Blotting Substrate Kit (Tanon, Shanghai, China).

### 2.9. Protein-Degradation Experiment

The Myc-*sc*PHD2 or pCMV-myc was co-transfected with Flag-*sc*HIF-1α into MFF-1 cells for 24 h, and then treated with cycloheximide (CHX; 100 μg/mL) for different time points. Cell extracts from each time point were resolved by SDS–polyacrylamide gel electrophoresis followed by Western blotting using Flag-tag, Myc-tag, and β-actin antibodies. The Myc-*sc*PHD2 or pCMV-myc was co-transfected with Flag-*sc*HIF-1α into MFF-1 cells for 24 h, and then treated with cycloheximide (CHX; 100 μg/mL) and MG132 (20 μM) for different time points. Cycloheximide (CHX) is an inhibitor of intracellular protein synthesis. MG132 is a proteasome inhibitor. At the end of the treatment, cells were harvested and lysed under denaturing condition. Western blots using antibodies against Flag-tag and Myc-tag were used to test *sc*HIF-1α and *sc*PHD2 amounts.

### 2.10. Statistical Analysis

All data analyses were carried out using SPSS 20.0 and all the experimental data were subjected to one-way ANOVA (one-way analysis of variance). For all analyses, significance was set at the 0.05 threshold (* *p* < 0.05; ** *p* < 0.01; ns represent not significant). All data are expressed as the mean ± standard deviation (SD).

## 3. Results

### 3.1. Molecular Characteristics of scPHD2

We performed PCR reactions with primers (Table 1) to clone the sequences of *sc*PHD2 and obtained cDNA fragments. This sequence was confirmed by the BLAST program. The full-length cDNA of *sc*PHD2 was 1723 bp, including a 5′-untranslated region of 357 bp, a 3′-untranslated region of 295 bp, and a 1071 bp open reading frame encoding a protein of 356 amino acids. The deduced amino acid sequences of *sc*PHD2 contained a ZF-MYND domain (15–52 amino acid residue) and a prolyl 4-hydroxylase alpha subunit homolog domains (P4Hc domain, 155–324 amino acid residue), well matching those of *Homo sapiens* PHD2 (*hs*PHD2, Figure 1A). Phylogenetic tree results showed that the PHD proteins were clustered into three major groups, namely, PHD1, PHD2, and PHD3 (Figure 1B). Within the PHD2 cluster, *sc*PHD2 formed a cluster with Perciformes (fish) PHD2, which was supported by a high bootstrap value. This finding indicated a closer relationship between *sc*PHD2 and other Perciformes PHD2 than between *sc*PHD2 and other PHD2. Thus, *sc*PHD2 was conserved in vertebrate.

The protein structure of *sc*PHD2 predicted by the amino acid sequence is shown in Figure 1C. The pLDDT of *sc*PHD2 and *hs*PHD2 by Alphafold were 77.50 and 71.90, respectively. Most amino acid sites’ pLDDT values were greater than 90 in *sc*PHD2. Pymol v2.5.0 was used to compare the protein structure of predicted *sc*PHD2 and *hs*PHD2, and we found that the structure of *sc*PHD2 was highly similar to that of *hs*PHD2. Furthermore, the amino acid sites of 122–338 in *sc*PHD2 was highly consistent with the amino acid site of 173–389 in *hs*PHD2, with an RMSD of 0.391. 

Subcellular localization and tissue distribution are dominant physiological functional characteristics and closely related to function. As shown in Figure 1D, the green fluorescence represents the Myc-tagged *sc*PHD2 and aggregates in the cytoplasm of MFF-1 cells. The tissue distribution of *sc*PHD2 in mandarin fish was examined by RT-qPCR. The expression level of *sc*PHD2 was constitutively detected in all tested tissues, including muscle, fin, liver, gill, fat, brain, spleen, heart, intestine, hind kidney, blood, middle kidney, and head kidney (Figure 1E). *sc*PHD2 expression in blood, liver, and gill was higher than in the other tissues, indicating that *sc*PHD2 may be involved in oxygen metabolism under normal physiological conditions. 

### 3.2. scPHD2 Inhibited the HIF-1 Signaling Pathway

The pGL4-HREs-luc plasmid is an HIF-1 reaction element (HRE), which can be combined with HIF-1α in the DNA sequence. To investigate the role of *sc*PHD2 in the HIF-1 signaling pathway of mandarin fish, dual-luciferase reporter gene assays were conducted. As shown in Figure 2A, the relative level of HRE-luciferin significantly decreased after *sc*PHD2 overexpression, suggesting that *sc*PHD2 inhibited the HIF-1 signaling pathway. With decreased concentration of transiently transfected *sc*PHD2, the inhibitory effect on the HIF-1 signaling pathway also decreased (Figure 2B). To verify these observations, the relative transcription levels of the *scglut-1*, *scvegf*, and *scldha* genes (*sc*HIF-1 signaling pathway downstream) were detected using RT-qPCR. Results showed that the relative transcription levels of the *scvegf*, *scldha*, and *scglut-1* genes significantly decreased after *sc*PHD2 overexpressed in cells. (Figure 2C–E). These results suggest that *sc*PHD2 could inhibit the *sc*HIF-1 signaling pathway in MFF-1 cells.

### 3.3. scPHD2 Interacted with scHIF-1α

To verify the molecular mechanism of how *sc*PHD2 negatively regulated the HIF-1 signaling pathway, the interaction between *sc*PHD2 and *sc*HIF-1α were tested by Co-IP analysis. Cells were transiently co-transfected with the vectors encoding Myc-*sc*PHD2 and Flag-*sc*HIF-1α. At 48 h post-transfection, cells were harvested and lysed. Equal amounts of protein were incubated with Myc/Flag-Sepharose beads and then analyzed via Western blot using the anti-Myc/anti-Flag antibodies. As shown in Figure 3A, *sc*PHD2 could precipitate with *sc*HIF-1α, and *sc*HIF-1α could also precipitate with *sc*PHD2 in vitro, suggesting that *sc*PHD2 interacted directly with *sc*HIF-1α. Furthermore, the co-localization signal as revealed by confocal microscopy proved the interaction. As shown in Figure 3B, *sc*PHD2 aggregated in the cytoplasm of transfected cells, but when *sc*PHD2 and *sc*HIF-1α were co-transfected into MFF-1 cells, *sc*PHD2 aggregated in the nucleus and the cytoplasm. These results suggest that *sc*PHD2 could interact with *sc*HIF-1α in MFF-1 cells. Accordingly, we hypothesized that the molecular mechanism of *sc*PHD2 negatively regulated the HIF-1 signaling pathway in MFF-1 cells, possibly through the interaction between *sc*PHD2 and *sc*HIF-1α proteins.

### 3.4. scPHD2 Promoted the Degradation of scHIF-1α and Its Degradation Pathway

A major function of PHD2 protein is hydroxylation, which leads to HIF-1α degradation. Given that the results confirmed an interaction between *sc*PHD2 and *sc*HIF-1α, we subsequently determined whether *sc*PHD2 could regulate the protein level of *sc*HIF-1α protein. Cells were treated with CHX to inhibit protein biosynthesis, and the protein extracts obtained at indicated time points were analyzed. We found that *sc*PHD2 overexpression profoundly decreased the protein level of *sc*HIF-1α (Figure 4A,B). Thus, *sc*PHD2 mediated the degradation of *sc*HIF-1α protein in MFF-1 cells. Furthermore, the effect of *sc*PHD2 on *sc*HIF-1α could be blocked by the proteasome inhibitor MG132 (Figure 4C,D), suggesting that *sc*PHD2 promoted *sc*HIF-1α degradation through the proteasome-degradation pathway. With decreased concentration of transiently transfected *sc*PHD2, the degradation effect on *sc*HIF-1α protein level also decreased (Figure 4E,F). Collectively, these results indicate that *sc*PHD2 promoted *sc*HIF-1α degradation through a proteasome-dependent manner.

## 4. Discussion

Hypoxic stress often occurs in aquaculture environments. Many farmed fishes, especially those in early developmental stages, are exposed to anoxic environments due to high density, excessive feeding, and inappropriate management [39]. The HIF-1 pathway has been extensively studied in model organisms, such as *Drosophila melanogaster*, *Caenorhabditis elegans*, and mammals [40,41,42,43,44]. Conversely, little is known about the function of this pathway in terms of its key importance in tolerating hypoxia in fish [45,46,47]. The PHD2 gene has only been cloned and identified with different expression patterns from *Megalobrama amblycephala*, *Sillago sihama,* and *Hypophthalmichthys molitrix* under hypoxia conditions [48,49,50], but the role of the HIF-1 pathway in the mandarin fish remains unknown. Thus, studying the responses and adaptive mechanisms to hypoxia challenge in mandarin fish is necessary.

The present research described molecular characterization of *sc*PHD2. The predicted amino acid sequence of *sc*PHD2 showed high homology with other vertebrates, especially with Perciformes (fish) PHD2. It had similar functional domains to other EGLN family members, including a ZF-MYND domain, a 2OG-Fe (II) oxygenase superfamily domain, and a P4Hc domain [7,13]. The 2OG-Fe (II) oxygenase superfamily domain is crucial to the regulation of hypoxia-inducible transcription factors, and it is a characterizing domain of PHDs [7,13,51]. The P4Hc domain catalyzes the proline hydroxylation of collagen to form 4-hydroxyproline, which regulates the hypoxic response by HIF-1α hydroxylation [52,53,54]. Accordingly, the conserved domains suggest that the PHD2 protein of mandarin fish have similar biochemical functions with other species [55,56]. However, the ZF-MYND domain is unique to PHD2 protein in the PHD family and is absent in PHD1 and PHD3 in vertebrates [13,57]. ZF-MYND has extensive evolutionary conservation, and in other proteins, it usually acts as a domain interacting with other proteins [14,58,59]. According to the conserved amino acids and domain structure, PHD2 is the most primitive form in the PHD homologous family of metazoans [13,54,60]. Phylogenetic analysis suggests that PHD1, PHD2, and PHD3 in different species cluster into a large clade, confirming that PHDs are relatively highly conserved in their coding sequences amongst vertebrates [61].

Under the condition of normal oxygen, the expression of *sc*PHD2 mRNAs in various tissues of mandarin fish was detected by RT-qPCR. Results showed that the *sc*PHD2 mRNA was expressed in all tissues, but the amount of expression differed. The *sc*PHD2 mRNA was highly expressed in blood, liver, and gill. The reason may be that *sc*PHD2 was more sensitive to changes in oxygen concentration in the blood and liver and played an important role in the adaptive response to hypoxia. Our results are consistent with previous studies on *Megalobrama amblycephala* and *Hypophthalmichthys molitrix. Ma*PHD2 has been found to be ubiquitously expressed in all detected tissues, with the highest level in peripheral blood, followed by brain, heart, and gill [48]. The mRNA level of *Hm*PHD2 was higher in gill and muscle [51]. However, our data are inconsistent with the results in mammals, in which PHD2 is most highly expressed in heart tissue, followed by brain and kidney [62]. Each PHD reportedly shows a different hydroxylation preference for HIF-αs in mammals [63]. The primary effect of PHD2 is primarily regulating the elevation of HIF-1α in normoxia, and PHD3 appears to contribute more substantially to the regulation of HIF-2α than HIF-1α because in most cells, PHD2 is essentially the most abundant HIF prolyl hydroxylase under these conditions [60]. However, PHD1 and PHD3 are involved in the regulatory system, and for PHD3, this contribution may be as much or more than that of PHD2 under the right conditions.

To understand how PHD2 regulated HIF-1α under normoxia in mandarin fish, dual-luciferase reporter gene assays were conducted. Results showed that *sc*PHD2 can inhibit the HIF-1 pathway under normoxia. The Co-IP experiment further proved that *sc*PHD2 interacted with *sc*HIF-1α, so we concluded that the molecular mechanism of *sc*PHD2 negatively regulated the HIF-1 signaling pathway, possibly through the interaction between *sc*PHD2 and *sc*HIF-1α proteins. Finally, the protein degradation experiment showed that *sc*PHD2 primarily promoted *sc*HIF-1α degradation through the proteasome degradation pathway. Our results are consistent with those of previous studies. In other studies, siRNA experiments demonstrated that PHD2 promotes HIF-1α hydroxylation and controls HIF-1α levels under normoxic conditions, whereas PHD1 and PHD3 do not promote HIF-1α hydroxylation in vivo [60]. Furthermore, the oxygen-dependent nuclear ubiquitination of HIF-α has been shown to be prevented by the inhibition of the HIF-specific prolyl hydroxylase, suggesting that the nuclear ubiquitination of HIF-α requires the nuclear prolyl hydroxylation of PHD protein [64]. Compared with PHD1 and PHD3, PHD2 is considered to be a rate-limiting enzyme that controls HIF-1α in low normoxic levels [60].

In conclusion, we cloned *sc*PHD2 and identified its effect on the HIF-1 signaling pathway. This study shed light on the regulatory functions of PHD2 under normoxia, thereby providing a reference for subsequent studies on the molecular mechanism of hypoxia adaptation in mandarin fish.

## 5. Conclusions

The full-length cDNA of *sc*PHD2 was 1723 bp and contained 1071 bp open reading frames encoding a protein of 356 amino acids. Amino acid sequence analysis showed that *sc*PHD2 had the conserved P4Hc domain at its C-terminus. Meanwhile, phylogeny evaluation revealed that *sc*PHD2 was more closely related to other Perciformes PHD2 than other PHD2. Tissue distribution results revealed that *scphd2* gene was expressed in all tissues tested, with the highest expressional level of *scphd2* in blood and liver. Subcellular-localization analysis showed that *sc*PHD2 was translocated into the cytoplasm. *sc*PHD2 overexpression could inhibit the HIF-1 signaling pathway. Co-immunoprecipitation analysis showed that *sc*PHD2 could interact with *sc*HIF-1α to promote *sc*HIF-1α degradation through the proteasome degradation pathway. This study advanced our understanding of how the HIF-1 signaling pathway is regulated by *sc*PHD2 and will help us to understand the molecular mechanisms underlying hypoxia adaptation in teleost fish.

## Figures and Tables

**Figure 1 animals-13-01556-f001:**
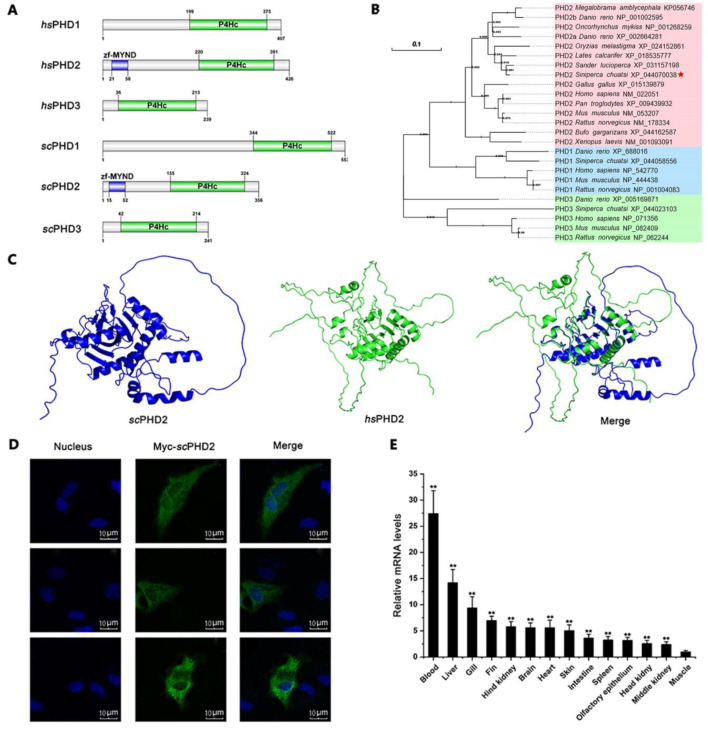
Molecular characteristics of *sc*PHD2. (**A**) Domain organization of *hs*PHD and *sc*PHD. *sc*PHD2 contains a ZF-MYND domain (15–52 amino acid residue) and a P4Hc domain (155–324 amino acid residue), well matching those of *hs*PHD2. (**B**) Phylogenetic tree of PHD proteins from various species. The values at the forks indicate the percentage of trees in which this grouping occurred after bootstrapping. The name of the species and its GenBank accession number are shown in the picture. Mandarin Fish is highlighted with a red asterisk. (**C**) Three-dimensional structure prediction of *sc*PHD2 and *hs*PHD2. The *sc*PHD2 protein and the *hs*PHD2 protein structures were predicted by applying the homology modeling technique in Alphafold v2.3.0. (**D**) Subcellular localization of *sc*PHD2 in MFF-1 cells. The nucleus was stained with Hoechst 33342, and fluorescent signals were observed under a fluorescence microscope. (**E**) Transcription levels of the *scphd2* gene in various tissues from healthy mandarin fish. The *β*-actin gene served as an internal control to calibrate the cDNA template for all samples. The *y*-axis represents the relative mRNA expression. Each bar represents the mean ± SD of triplicate samples. Data are representative of three independent experiments. ** *p* < 0.01.

**Figure 2 animals-13-01556-f002:**
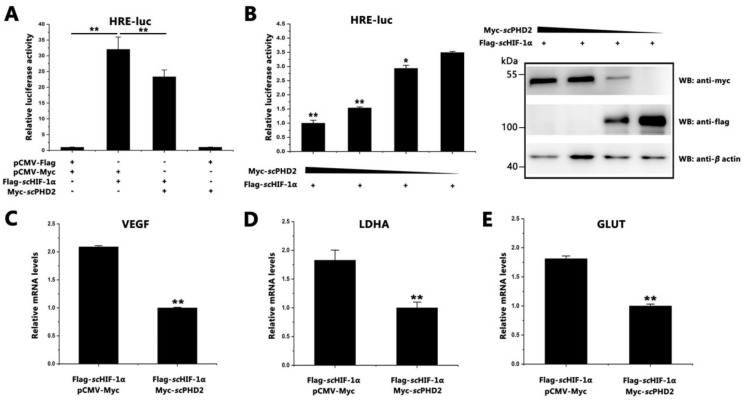
*sc*PHD2 inhibited the HIF-1 signaling pathway. (**A**) The *sc*HIF-1 signaling pathway was inhibited by *sc*PHD2. When the MFF-1 cells reached approximately 70% confluency, they were used for transfection. Four treatments were designed: 0.2 μg pGL4-HREs-luc + 0.2 μg pCMV-Flag + 0.2 μg pCMV-Myc + 0.02 μg pRT-TK; 0.2 μg pGL4-HREs-luc + 0.2 μg Flag-*sc*HIF-1α + 0.2 μg pCMV-Myc + 0.02 μg pRT-TK; 0.2 μg pGL4-HREs-luc + 0.2 μg Flag-*sc*HIF-1α + 0.2 μg Myc-*sc*PHD2 + 0.02 μg pRT-TK; 0.2 μg pGL4-HREs-luc + 0.2 μg pCMV-Flag + 0.2 μg Myc-*sc*PHD2 + 0.02 μg pRT-TK. Luciferase activity was measured after 24 h. Firefly luciferase activity value was compared with *Renilla* luciferase activity. The *y*-axis represents the relative luciferase activities. Data are presented as the mean ± SD from three independent triplicated experiments. **, *p* < 0.01 versus the controls. (**B**) Increased *sc*PHD2 inhibited the *sc*HIF-1 signaling pathway in a dose-dependent manner. MFF-1 cells were transfected with either Flag-*sc*HIF-1α or increasing doses of Myc-*sc*PHD2, and the firefly luciferase activity value was compared with *Renilla* luciferase activity. Flag-*sc*HIF-1α and Myc-*sc*PHD2 expression were analyzed by Western blotting after 24 h of transfection (right image). Data are presented as the mean ± SD from three independent triplicate experiments. **, *p* < 0.01; *, *p* < 0.05 versus the controls. (**C**–**E**) Expression patterns of *scvegf*, *scldha*, and *scglut-1* genes determined by RT-qPCR after cells were overexpressed with *sc*HIF-1α and *sc*PHD2. The *β*-actin served as an internal control. The *y*-axis represents the relative mRNA expression. Asterisks above bars represent statistically significant differences among the control samples. **, *p* < 0.01.

**Figure 3 animals-13-01556-f003:**
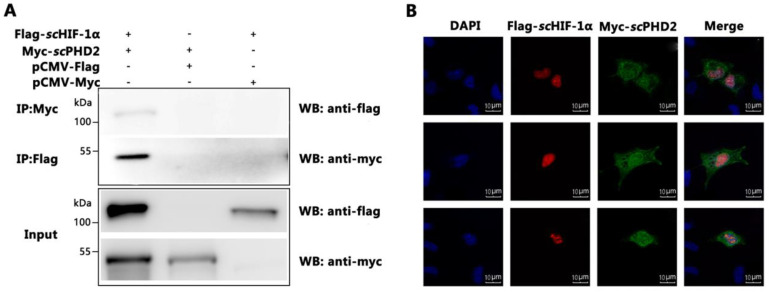
*sc*PHD2 interacted with *sc*HIF-1α. (**A**) The interaction of *sc*PHD2 with *sc*HIF-1α was determined by Co-IP assay. MFF-1 cells co-expressed Flag-*sc*HIF-1α/Myc-*sc*PHD2 (lane 1), Flag-tag/Myc-*sc*PHD2 (lane 2), and Flag-*sc*HIF-1α/Myc-tag (lane 3). Immunoprecipitation of Flag-*sc*HIF-1α with anti-Flag and IP of Myc-*sc*PHD2 using anti-Myc. (**B**) The co-localization signal as revealed by confocal microscopy proved the interaction of *sc*PHD2 with *sc*HIF-1α. Subcellular localization of *sc*PHD2 in MFF-1 cells transfected with Myc-*sc*PHD2 and *sc*HIF-1α in MFF-1 cells transfected with Flag-*sc*HIF-1α. After allowing the cells to adhere for 48 h on 24-well plates, the nucleus was stained with Hoechst 33342. Fluorescent signals were observed under a fluorescence microscope.

**Figure 4 animals-13-01556-f004:**
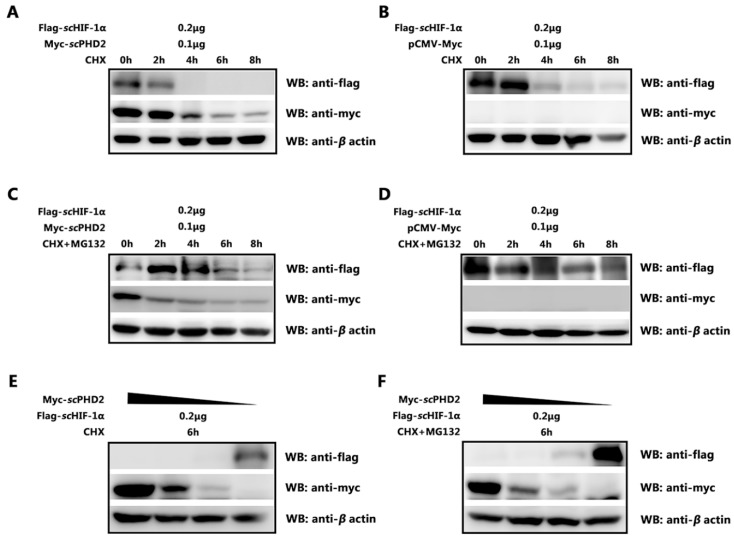
*sc*PHD2 promoted the degradation of *sc*HIF-1α and its degradation pathway. (**A**,**B**) After overexpressing Flag-*sc*HIF-1α/Myc-*sc*PHD2 or Flag-*sc*HIF-1α/Myc-tag in MFF-1 cells, treatment with CHX at different time points showed that *sc*PHD2 accelerated *sc*HIF-1α degradation. β-Actin served as an internal reference. (**C**,**D**) After overexpressing Flag-*sc*HIF-1α/Myc-*sc*PHD2 or Flag-*sc*HIF-1α/Myc-tag in MFF-1 cells, treatment with CHX and MG132 at different time points showed that the effect of *sc*PHD2 on *sc*HIF-1α could be blocked by the proteasome inhibitor MG132. β-Actin served as an internal reference. (**E**,**F**) At 6 h post-treatment with CHX or CHX + MG132, the degradation effect on *sc*HIF-1α protein level decreased with decreased concentration of transiently transfected *sc*PHD2. β-actin served as an internal reference.

**Table 1 animals-13-01556-t001:** Primers used for cDNA cloning of *sc*PHD2-conserved regions.

Name (For Initial PCR)	Sequences
5′ RACE for *sc*PHD2-F	5′–CTAATAGCACTCACTATAGGGCAAGCAGTGGTATCAACGCAGAGT–3′
5′ RACE for *sc*PHD2-R	5′–ACACTGTGATGGCATACCTGGTGGC–3′
3′ RACE for *sc*PHD2-F	5′–GGACTACGAGGCACCGGAGATAA–3′
3′ RACE for *sc*PHD2-R	5′–ACTCTGCGTTGATACCACTGCTTGCCCTATAGTGAGTGCTATTAG–3′
*sc*PHD2-F1	5′–ATGGAGAAGCAGCAGAGCGATTTGGAC–3′
*sc*PHD2-R1	5′–CTAGCTGGGATCTGATGGTTTGCCGA–3′

**Table 2 animals-13-01556-t002:** Primers used for quantitative PCR.

Genes	Primers	Sequences	Primer Efficiency
*scvegf*	Forward Reverse	5′–ACCGAAGGAAACAGAAAGAGG–3′ 5′–CAGGACGGGATGAAGATGTG–3′	0.98
*scldha*	Forward Reverse	5′–GGTCTTCCTGAGCATCCCTT–3′ 5′–TTCTCCTCTTCGGGCTTCA–3′	0.98
*scglut 1*	Forward Reverse	5′–GGTTTATTGTGGCAGAGTTGTT–3′ 5′–CCCACTATGAAGTTGGCAGTC–3′	0.97
*β-actin*	Forward Reverse	5′–CCCTCTGAACCCCAAAGCCA–3′ 5′–CAGCCTGGATGGCAACGTACA–3′	0.96
*scphd2*	Forward Reverse	5′–ACACCGCCACATCTAACG–3′ 5′–GTGCAGGGATTTGACATTCT–3′	0.98

## Data Availability

The data presented in this study are available on request from the authors.

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
