# Peer review of "Molecular Characterization and Functional Analysis of Hypoxia-Responsive Factor Prolyl Hydroxylase Domain 2 in Mandarin Fish (Siniperca chuatsi)"

_animals, 2023, doi:10.3390/ani13091556_

Round 1

Reviewer 1 Report

PHDs are critical proteins that regulate HIF-1 signaling pathway in response to hypoxia, especially regulate the stability of HIF-1α. This manuscript investigated the roles scPHD2 of mandarin fish (Siniperca chuatsi) in HIF-1 signaling pathway, scPHD2 could interact with scHIF-1α and promote scHIF-1α degradation through the proteasome-degradation pathway. This work would be helpful to understand the roles HIF-1 signaling pathway in response to hypoxia in teleost fish.

There are some minor comments/suggestions:

1. Line 59, two conjunctions cannot be used together.

2. Line 73, this sentence does not read well.

3. Lines 164, Table 2, please include primer efficiency numbers in qPCR table.

4. Line 265, this sentence does not read well.

5. Figure 2A, need to mark the significance of those two groups.

6. Figure 2A-B, The plus sign in the figure needs to be enlarged to allow observation of the description.

7. Line 180/292, The description of the experimental method should be consistent.

Reviewer 2 Report

In this study, the PHD2 of mandarin fish was cloned and investigated its interaction with HIF-1α. The paper showed useful information to understand the molecular mechanisms of hypoxia adaptation in mandarin fish. While, there are some questions:

1. The animals were purchased from a farm. How to judge the animals were healthy?

2. Is the information of RT-qPCR primers for scphd2 missing in the Table 2?

3. There should be a result of statistical analysis in the Fig1-E.

4. Is there missing part of annotation for Fig4?

Reviewer 3 Report

The authors investigated the Molecular Characterization and Functional Analysis of Hypoxia-Responsive Factor Prolyl Hydroxylase Domain in Man-3 darin Fish Siniperca chuatsi. This manuscript (MS) was clearly written and easy to understand. This work can help the sustainability of this species farming as few studies have been done on this topic. However, some minor issues significantly compromised the quality of this MS.

However, I have touched on some more points that can contribute to the improvement of this MS.

·       Here and throughout the MS, please first mention the common name plus scientific name, and for the rest of the MS, just report the common name.

·       Line 23, revise.

·       Line 394, please improve the conclusion section.

·       Here and elsewhere, report P uppercase and italic (P<0.05).

·       Throughout the MS, if there is no significant difference, no need to report P-value.

·       Please reorder the keywords alphabetically and capitalize each word.

·       Please write the abstract more numerically about the results. You can do it by adding their numbers in parentheses.

·       Please update the introduction with recent works as many studies are available from the last two years, which were not included in this section.

·       Please mention the novelty of your work in the last paragraph of the introduction.

·       For each analysis, please clarify how many fish were taken.

·       As a general comment: please focus on fish as hips of references and studies are available, and no need to cite other vertebrates.

·       Although you wrote this section well, you can still improve it by answering these questions and annotating them into the discussion section. Why were these results observed? Discuss more possible reasons.

Tables and Figures

•            Please explain a little bit about your experimental treatments, per each Table and Figure. Each Table and figure should represent enough information separately from the text.

•            Double-check the units and titles of all Tables.

•            Please mention in the footnote of all Tables which kind of statistical method you used for comparing the means.

When revising your manuscript, please consider all issues mentioned in the reviewers' comments carefully with clear outlines for every change made in response to their comments including suitable rebuttals for any comments you deem inappropriate. Please itemize your response to each review comment, and highlight the revised at re-submission.

Best regards

Reviewer 4 Report

The theme of the article is very important in understanding the mechanism of hypoxia, one of the fish management challenges in aquaculture. The novelty of the work in this manuscript will grab the attention of researchers who are interested in studying mechanisms of hypoxia in Mandarin fish to read it. However, some of the methods need to be properly described, and the language needs a moderate polish by a native English speaker. 

Some minor comments are indicated below:

Ttil: Could you please put ‘Siniperca chuatsi’ in Parentheses 

2. Materials and methods

-          Please, combine the transfection description in lines 92 – 97 with ‘ 2.5. Dual-luciferase reporter gene assays’ described in lines 130 – 140

-          Line 99: Please change to "was obtained". 

-          Was the MFF-1 cell characterized before? Any publications? Reference?

-          Line 102: what instruction did you follow to isolate total RNA? Please, use a reference or describe what you did.

-          Line 116: Please, change SMART link to the correct link, the one you added does not work.

-          2.5. Dual-luciferase reporter gene assay needs to be described properly

-          Line 130: Please, indicate the aim of using the dual luciferase reporter gene assay as you did in 2.7 Indirect immunofluorescence assay (IFA).

-          Line 130: What plasmid? And why different amounts? Was that for optimization of transfection? Please elaborate and change ‘amounts’ to ‘concentrations’.

-          Line 132: What is the source of the plasmids you used?

-          Lines 133 and 134 and lines 265 -266: I think it is not the Flag-scHIF-1α or Myc-scPHD2 co-transfetced with Flag-scHIF-1α that are positive control and experimental groups, but rather the cells that were transfected with these plasmids, right?

-          Cells transfected with control and experimental groups or reporters?

-          Could you please provide a sketch describing 2.5. Dual-luciferase reporter gene assays.

-          2.6. Real-time quantitative PCR (RT-qPCR): please indicate why RT-qPCR was used in the first line of the respected paragraph as you did in 2.7 Indirect immunofluorescence assay (IFA).

-          Lines 146: Expression analysis of which genes ??

-          The authors mentioned that the primers used in real-time are shown in table 2 a couple of times (in lines 145 and 151) please avoid redundancy.

-          How many reference genes were used in RT-qPCR experiment? Was B-actin suitable for all tissues, can you show the crude data for B-actin in different tissues?

-          Table 2 please mention the primers’ efficiency and Accession numbers of the different genes analyzed.

-          Line 163: Please, change ‘Endo-free plasmids’ to endotoxin-free plasmids, the former is a name of a kit.

-          Line 174: what Plasmids?

-          Line 178: analyzed or treated?

-          You do not need to mention how the cells were transfected (EZ3000 Plus protocol) in each subsection of the methods section. It is enough to describe it properly once under one subtitle and refer to it every time you want to mention it. However, it is important to describe which plasmids were used for transfection in each subtitle in the methods section.

-          Line 193: which plasmids? ‘Various’ does not give an indication of which plasmids were used.

-          Line 193: Was the EZ3000 Plus used for 24 hours

-          Line 194: followed by cycloheximide treatment? Please spell it out!

-          Line 194:   The authors mentioned in this line «for various times as indicated in the Results». Please, make everything clear in the methods, before you go into the results. What do you mean by various times? You mean time points? Please change!

-          Line 196 and 199: «…..using FLAG-tag, Myc-tag or B-actin Antibodies, not Tag-FALG, Tag-Myc». Please, change!

-          Line 197: were the cells transfected and then treated with MG132, or what? Please clarify and reformulate the paragraph. It is also not clear from the writing why MG132 was used, please elaborate!

Results

-          Line 202: What do you use ‘sequences’,  did you get more than one sequence using one primer pair in table 1?

-          Line 226: Transcription level, not ‘transcriptions levels.

-          Figure 1E: which blood cell type was analyzed for the expression of scPHD2, RBC or WBC?

-          Line 247 and 269 the author mentioned that data are represented as mean ± SEM, while in line 159, it was mentioned that ‘all data are expressed as the 158 mean ± standard deviation (SD)’. Which of them is correct? And the author means by mean ± SE in line 274.

-          Line 250: What the authors mean by  «The pGL4-HREs-luc plasmid is an HRE ……».

-          Line 250: Please, change to be combined with.

-          Line 267: by « The luciferase activity» do you mean Firefly luciferase activity or total luciferase activity?

-          In figure 3A: in the INPUT panel, by ‘Input’ you mean the cell lysate without beads? If so, how anti-flag Ab recognize a band in lysate containing Myc-scPHD2 and control Flag control plasmid on lane 3, and how does using anti-myc Ab showed a band in cell lysate containing Flag-scHIF-1a and Myc control plasmid in lane 2?

- In the caption of Figure 4 Line 322 is there something missing before .".....and (B)"?

Language:

line 333: please change to 'early-developmental stages', ' exposed to"

Line 335: Drosophila melanogaster, 335 and Caenorhabditis elegans are not mammals, please change!

Line 337: remove 'important'

Line 337: please change to ' of its key importance in tolerating hypoxia in fish'

_ Remove  line 338.

- Please remove all  reference to Figures from the Discussion.

- Line 365: I did not get the meaning of 'Oxygen perception', could you please change that.

- Line 371: ' HIF-αs' ? what is the 's' in the end? Could you please start the sentence in Line 372 with 'In mammals,.......'.

- Line 386: The authors started the sentence with 'Their', who the authors are referring to by 'Their'?

- Line 388: please make ' in vivo' italicized'. 

- Line 391: Please remove 'Therefore'

- Line 394: I do not think the authors identified PHD2 function, could you please reformulate to something like. '......... and demonstrated its effect on HIF-1 pathway...'. 

- Line 395: Please change 'great significance' to 'This study shed light on the regulatory functions of PHD2 under normoxia.

Language:

Please pass the manuscript to an English native speaker to polish the language.
